# Earthquake Response for Students with Different Severe Degrees of Disabilities: An Investigation of the Special Education Classes in Primary Schools in Taipei

**DOI:** 10.3390/ijerph19148750

**Published:** 2022-07-18

**Authors:** Yung-Fang Chen, Kuo-Chen Ma, Mei-Hsiang Lee, Mo-Hsiung Chuang

**Affiliations:** 1School of Energy, Construction and Environment, Coventry University, Coventry CV1 5FB, UK; aa4106@coventry.ac.uk; 2Department of Urban Planning and Disaster Management, Ming Chuan University, Taoyuan 333, Taiwan; bigbear@mail.mcu.edu.tw; 3Hulu Elementary School, Taipei 111, Taiwan; carol5705@gmail.com

**Keywords:** special education, disabilities, comprehensive school safety framework, earthquake response, DRR curricula, evacuation

## Abstract

Taipei has been ranked as the most vulnerable city to a wider combination of risks. Although the Special Education Law addresses the consideration of disaster preparedness for students with disabilities enrolling in Special Education Schools, more attention needs to be given to the far larger number of students with disabilities enrolling in normal schools. These schools need to consider the care for students with different types and severe degrees of disabilities. The aim of the research is to investigate challenges of earthquake preparation and response for students with different severe degrees of disabilities who enrol in the special education classes in general primary schools. The objectives of the research include the following: (1) investigating the challenges and requirements for support of students with different severe degrees of disabilities; (2) examining the need and support for students with different degrees of disabilities during the earthquake response process; and (3) exploring the best practice in the curriculum building for students with different severity of disabilities. The purposive sampling was used to select four primary schools in Taipei as participant groups in the research. The research team used semi-structured interviews to interview eight participants: one special education class teacher and one administrator of each school were invited. Findings include the following: (1) ensuring the appropriate design of physical environment for earthquake response in schools, including rapid response, evacuation, and assembly points for students with different severe degrees of disabilities; (2) proposing suitable staff to student ratio to be planned for the response phase; (3) identifying the appropriate individualised curriculum and learning objectives to suit students with different severe degrees of disabilities.

## 1. Introduction

Taipei has encountered very frequent earthquakes, on average 2200 per year, including more than 200 strong enough to be felt without instruments [1]. More significantly, Taipei has experienced nearly 100 major earthquakes since 1900, each toppling buildings, half resulting in loss of life; in particular, the Ji-ji earthquake of 1999 resulted in more than 2400 deaths, 11,300 injuries, 100,000 buildings collapsed, and 700 school buildings damaged [2]. School safety has drawn attention after this earthquake since children in Taipei have very long school days and it is highly possible that they will face any of the disaster risks while they are in school.

From 2003 to the present, four phases of implementation have been introduced by the Ministry of Education to ensure students’ safety in schools. The first phase started with the “Disaster Management Technology Education Personnel Cultivation Pilot Programme” [2]: a disaster education curriculum was developed; teacher-training programmes were designed; and campus infrastructure safety was improved. The second phase was the “Disaster Management Technology Education Experiment & Research Programme” [3], which started to develop teaching materials from primary schools to continuing professional development training. According to the “School Disaster Management Network Establishment & Experiment Programme” [4], the third phase was to promote early warning systems in schools located in high-risk areas. The fourth phase, “School Disaster Risk Prevention & Reduction and Climate Change Education Adjustment & Enhancement Programme” [5], emphasised the importance of localised disaster risk reduction education and promoted knowledge and expertise. The “Disaster Risk Education White Paper” [6] laid down that schools should use the framework of the disaster management cycle: mitigation, preparation, response, and recovery, when establishing their emergency plans. The mitigation phase focused on the enhancement of school infrastructures, for example, buildings and learning facilities. The preparation phase highlighted the need for conducting regular exercises and emergency response drills. Teacher training in hazard awareness was also recognised as part of preparedness.

During this period, some considerations were paid to enhance children’s safety in special education schools. Examples included “Trial of disaster prevention plan and exercises for preschool and special education schools” [2], “Guidance for writing disaster management plans for special education schools”, and “Special education school campus disaster management plans” [3]. Although progress in the disaster risk reduction in the special education schools was observed, it is suggested that more effort should be made in the areas of evacuation procedures, staff to student ratio, and community support should be included in the disaster management plans. Further improvements for disaster risk reduction in special education schools were recommended: (1) establishing the operation procedures for students with disabilities; (2) enhancing special education teachers’ capacity and knowledge in DRR; and (3) developing suitable curriculum and learning tools for children with disabilities [7].

Based on the recommendations of previous projects, the research aimed to further investigate challenges of earthquake preparation and response for students with different severe degrees of disabilities who enrol in the special education classes in general primary schools. The objectives of the research include (1) investigating the challenges and requirements for support of students with different severe degrees of disabilities; (2) examining the need and support for students with different degrees of disabilities during the earthquake response process; and (3) providing recommendations and best practice for schools with similar settings to improve their support for students with disabilities to respond to earthquakes.

### 1.1. School Emergency Preparedness for Children with Disabilities

The growing number of natural hazards and their impact has focused attention on the area of emergency preparedness and disaster risk reduction. Many researchers have raised concerns about the risk that children face during disasters because they are the biggest sector of populations affected by disasters and they are the most vulnerable [8]. Evidence shows that about 10% of children worldwide have a form of disability and more than 3% of these children annually are affected by disasters. Children with disabilities are particularly vulnerable because they might have mobility difficulties or underlying medical conditions. They are also dependent on adults as their physical, emotional, and cognitive abilities for safety are still developing. They are not yet able to protect against emergencies and disasters [9,10]. In sudden-onset disasters, children with disabilities usually find it difficult to evacuate and might expose themselves to risks during the evacuation process [11]. Shelters might not have adequate support for these children due to shortage of experienced staff and appropriate equipment [12].

Research shows that most schools are underprepared for children with disabilities [8,11,13]. School emergency response plans should take account of the requirements of students with disabilities [14], including being part of a group during evacuation, their ability to interpret information regarding the guidance of response, and to adapt to a change of environment such as light change and noise [13]. Additional educational and physical support is required for those children while in school, and they should receive direct supervision from a special educator [8]. Different types of infrastructure and tools should be provided to support students with different types of disabilities during the emergency response phase [15]. Special education teachers could help students develop resilience and improve their response to disasters by developing individualised plans [11,13,16]. In addition, it is necessary to understand how children with disabilities access and use resources; the perception, skills, and strategies they adopt in facing a disaster; and to identify when and how to provide support [8]. Recommendations also include the involvement of parents and communities in the emergency response [17]. The major factors that prevent schools from effective planning and preparedness for students with disabilities have been identified, including: “(1) low school budget resulting in a lack of appropriate facilities, inaccessible school buildings, high pupil to teacher ratio, limited support for children with disabilities; (2) teachers have inadequate training in supporting children with disabilities during emergencies, and (3) limited awareness of disability among teachers and school staff” [18].

However, the lack of a standardised definition of disabilities and the complexity of measuring disabilities mean that it is more challenging to prepare children with disabilities who are not enrolled in the special education schools. There is no such standardised definition of disabilities [19]. It is argued that the complexity of measuring disabilities results from having no single definition that can apply to all cultural/geographical contexts; in addition, the perception about disabilities could also influence the results of the data gathering [19]. WHO and ICF utilised a “bio-psycho-social model” to define people with disabilities since it is a practical compromise between medical and social models of definition. It is agreed that disability refers to problems with human functioning in three interconnected areas: impairment, activity limitations, and participation restrictions [20]. ICF also stressed the interlink of environmental and personal factors. The severity of disabilities also influences the competence and capacity of students with disabilities [21]. During an emergency, carers normally are more needed to support students with more severe degrees of disabilities. Hence, it was suggested that identifying the cognitive ability and physical skills of students would be helpful for emergency planning; for example, students with multiple disabilities who have both moderate to severe physical impairments and severe mental disability require more support from teachers/staff as they have difficulties in movement and understanding instructions. Based on the cognitive ability and physical skills of students, the research has distinguished four levels of severe degrees of disabilities:Level 1: Able to respond and evacuate on their own—they can follow instructions and have a sense of space.Level 2: Need some help with response and evacuation—they have no problems in movement, they can also follow the example of others; however, they will need more support and guidance in the evacuation and assembly phase.Level 3: Poor response and evacuation ability: they might have difficulties in movement, need support from teachers/carers. They have difficulties in understanding earthquake knowledge and cannot take appropriate action; they need one-to-one support from a teacher/carer.Level 4: No ability to respond and evacuate by themselves. These students rely on wheelchairs or other types of support equipment to move and to arrive at the assembly point. One to two teachers/carers are required for support.

### 1.2. Education for Children with Disabilities in Taipei

Children in Taipei receive compulsory education between 6 and 15 years old in primary and junior high schools [22]. To ensure all students receive appropriate and suitable support for receiving education, the Special Education Law [23] provides multiple entrance routes for students with disabilities, including special education schools and general schools with special education classes. An individualized education plan for each and every special needs student should be established. Disability is defined as “physiological or psychological disorders, assessed and diagnosed by professionals to be in need of special education” and 13 types of disability are identified: intellectual disabilities, visual impairments, hearing impairments, communication disorders, physical impairments, cerebral palsy, health impairments, severe emotional disorders, learning disabilities, severe/multiple impairments, autism, developmental delays, and other disabilities [24].

Children with disabilities have four schooling options: home schooling, special education schools, special education classes in general schools or normal classes in a general school, depending on the type and severity of their disabilities. For example, the Ministry of Education [24] indicates that children with severe hearing, visual, learning, and multiple disabilities could enrol in the special education schools where supportive facilities and learning equipment are provided. Children with mental disorders, cerebral palsy, multiple disorders or autism and have severe cognitive impairment will be assigned to study in special education classes in general schools, and they can also choose to study at a special education school. Children with mild learning difficulties, autism, cerebral palsy, or multiple disorders could enrol in the class in a general school with special education support resources. Children who have no problem with cognition capacity but with emotional behaviour disorder, learning disability, language disorder, physical disorder, physical illness, and other disorders could enrol in the class in a general school. Students with mild visual and hearing disabilities can enrol in the normal class and will be supported by teaching assistants. According to the statistical data, there were, in total, 119,150 students with disabilities, of whom only 5953 students (5%) were enrolled in the 28 special education schools. The majority (95%) of students with disabilities go to general schools, either in a special education class or a normal class [25]. Although Taiwan has introduced a series of disaster risk reduction projects to enhance students’ safety at school, the lack of disaster risk reduction considerations for the 95% of students with disabilities attending special education classes in normal schools has been noted [26]. It is challenging to prepare disaster response for those students in the special education classes in a normal school, not only because students have a wide variety of disabilities, but also because schools need to integrate facilities and support for all their students. For example, Su [15] identified the requirements of school infrastructure for students with different types of disabilities during the evacuation phase (shown in Table 1).

More attention needs to be paid to students with disabilities enrolling in general schools. Apart from students without disabilities, these schools need to consider the care for students with disabilities. Some guidance was found to support students with different types of disabilities; however, students with different severe degrees of disabilities also required different levels of support.

## 2. Data Collection and Analysis Methods

The research team used semi-structured interviews and documentary review for data collection. The purposive sampling was used to select four (out of 17) primary schools running both normal and special education classes in Taipei as participant groups in the research. Students with intellectual disabilities, cerebral palsy, autism, and multiple disabilities are enrolled in these special education classes. One teacher and one administrator from each school were interviewed: one teacher who led the special education class and the administrator who was involved with school disaster management plans. The research team also used documentary review to support the validation of the results, including relevant government guidance, school disaster management plans, and research papers. Although it was identified that a quantitative method could display the facts of school safety arrangements, it is argued that the use of a qualitative method could further understand the need and gaps of the arrangements in schools for disabled children. Details of participant information are listed in Table 2.

To examine how to support students with different severe degrees of disabilities during the earthquake response phase, which includes in situ response, evacuation, and assembly phases, four levels of severe degrees of disabilities were distinguished by the research team for analysis:Level 1: Able to respond and evacuate on their own—they can follow instructions and have a sense of space.Level 2: Need some help with response and evacuation—they have no problems in movement, they can also follow the example of others; however, they will need more support and guidance in the evacuation and assembly phase.Level 3: Poor response and evacuation ability: they might have difficulties in movement, need support from teachers/carers. They have difficulties in understanding earthquake knowledge and cannot take appropriate action; they need one-to-one support from a teacher/carer.Level 4: No ability to respond and evacuate by themselves. These students rely on wheelchairs or other types of support equipment to move and to arrive at the assembly point. One to two teachers/carers are required for support.

The process of research and framework for analysis is listed in Figure 1.

## 3. Discussion

The earlier section identified the need to improve the arrangements to support students with disabilities. Schools should take into consideration individual students’ capacity and ability in the response, evacuation, and assembly phases in both physical environment and support staff arrangements; disaster management plans should include response time, classroom location, support resources, earthquake-resilient facilities, and the number of support staff. The research further explored and recommended measures to assist students with different severe degrees of disabilities. The section summarised the feedback from interviewees regarding how to improve the support for students with different severity of disabilities from the perspectives of school infrastructures, disaster management plan, and disaster risk reduction curriculum. Investigations also worked around the recommendations to support students with different severity of disabilities during in situ response, evacuation, and assembly phases. Areas for policy changes were also recommended.

### 3.1. Safe Learning Facilities—Challenges of Physical Environment

The Ministry of Education has enacted the Special education school establishment, change, closure, merger and staffing standards to provide guidance on the establishment of appropriate campus infrastructure for students with disabilities; for example, classrooms should be placed on the first floor; in addition, PE and daily skill classrooms should be provided to support teaching and learning. Schools have been tasked to establish the “School Building Safety Inspection Team” to ensure campus environment safety since 2010. Areas for inspection include campus building complying with the required building code; safe evaluation route; installation of fire alarm and fire safety equipment; and classroom facilities. This research focuses on the classroom facilities and evacuation route.

#### 3.1.1. In Situ Response—Resilient Classroom Environment

“Hide, drop, and hold on” is suggested to be the appropriate action to take to reduce injury and death during earthquakes; considerations for evacuation and assembly point are also required. Students with disabilities need to take a different approach for self-protection. In any case, classroom facilities play an important role for students’ safety. Areas for consideration included earthquake-proof shelves, wider doors, adhesive film on glass to prevent shattering, safe areas in the classroom, protective devices within easy reach, and space for wheelchairs.

The feedback from participants showed that earthquake-proof shelves are the most urgent items for students with any types and severities of disabilities. The next most important classroom facilities identified by participants included adhesive film on glass to prevent shattering and ensuring that protective devices are within easy reach. Participants also suggested that level-3 and level-4 students require safe areas in the classroom. “Teachers need to know where the comparable and safe areas in the classroom are” (Participant B4). “It is more challengeable to find a safe place in the daily skill classroom because there is no desk or other furniture” (Participant A3). It is suggested that teachers needed to move children from underneath pendant lights and ceiling fans to avoid falling hazards. To ensure the effectiveness of response, children with disabilities should always be provided with supportive tools such as head protectors during the normal school day. Finally, space for wheelchairs and wider doors should be designed for level-4 students.

#### 3.1.2. Evacuation and Assembly Phases

Four considerations should be included for the evacuation and assembly phases in the perspective of physical environment: what floor level the building the classroom was on, how long evacuation from the building would take, how uneven the floor was (important for all students, but critical for wheelchairs and other mobility aids), and how many obstacles there were on the evacuation route.

The location of the classrooms is important for evacuation because during this phase students with disabilities will need more time to get ready, so classrooms would preferably be located on lower floors. Most participants agreed that level-1 students can be situated on the second or third floors of the building; level-2 students can be on the second floor but with a short evacuation distance. Level-3 and level-4 students need to be situated on the first floor (ground level) and should be close to the assembly point. In the research, most schools have placed the classrooms on the first or second floor. Since Case C has more level-3 students, reconsideration of location of the classrooms is suggested.

In addition, evacuation routes need to be kept clear of obstacles and uneven surfaces, for example, “brooms, mops and other cleaning tools placed by the classroom doors could cause falling and tripping hazards to children” (Participant B1). Although difficulties and challenges were expressed, it was agreed that “the priority is to make sure that children are leaving the site; it does not matter how we make it happen, maybe push or carry them. During an exercise, we can make it perfect, but if the real situation happens, we need to make sure that children leave the site” (Participant B1). Case D adopted a two-point approach to bring students together after an earthquake; the complexity could be reduced by using one assembly point.

Apart from the evacuation routes, the research team also makes recommendations for the assembly points. All participants recommended that meeting away from basketball stands or other potentially dangerous constructions/building is essential. For level-1 and level-2 students, it is necessary to have a fixed assembly point. Having supportive volunteers and teachers nearby is also crucial for level-3 and level-4 students. It is suggested that Case B assigns a fixed assembly point.

### 3.2. School Disaster Management Plans: Earthquake Response

The Disaster Risk Reduction Education White Paper [6] laid down that schools should use the disaster management cycle: mitigation, preparation, response, and recovery, to establish emergency plans. To plan for earthquake response, schools are required to consider the timing of evacuation, evacuation route, allocation of refuge area, and assembly point in the disaster management plan [5]. To provide sufficient protection for children with disabilities, disaster management plans need to consider the challenges faced by children with disabilities and the required support from teachers and/or staff.

#### 3.2.1. Earthquake In Situ Response: A Need to Prioritise the Teacher/Staff Support According to Student Needs

Disaster management plans should take consideration of each children’s need to ensure sufficient teachers and staff are ready to help during an earthquake. “Drop, hide and hold on” is crucial during the in situ response phase [27]. Children with different types of disabilities should take alternative approaches with extra teacher support.

Children with wheelchairs, walkers, and canes should take alternative drop, hide, and hold-on positions, which is “to protect their heads, lower their body, and try to stay nearby strong tables/desks or pillars” (Participant B1). Students with multiple disabilities and who can move by themselves should “protect their heads” (Participant B3). Consideration should be given to those students with severe physical disabilities as it is challenging for them to take action by themselves: they will probably require help from teachers. “Some students have trouble with arm movements; under this circumstance, teachers will take them to a safer area” (Participant B3). However, “it is not possible to carry students from chairs and let them hide under the table; hence we need to make sure the head protectors are provided and are ready nearby” (Participant A2).

Teachers need to help students with wheelchairs or special supportive tools to hold on. “If they are sitting on a chair or wheelchair or special supportive tool, and are not able to drop and hide, they need to learn to hold on because if they become unstable/unsteady, they might have a fall and become in danger, so carers need to help to stabilise the wheelchairs” (Participant B2). “If they are not stabilised, they might have a fall during the shake, so we need carers to help to fix the wheels” (Participant B2). Given the extra support required, it seems that the current teacher-to-student ratio will not be sufficient in all cases.

In addition, teachers should protect themselves as a priority, for example, practice drop, hide, and hold on, or at least use the available head protectors. In addition, risk reduction and prevention warnings, for example, earthquake alerts, should be audible from the classroom; there should be sufficient response facilities, including bags, hats, cushions, and so on. Safety helmets for teachers should always be within reach. Teachers’ emergency grab bag should be distinguishable from normal household grab bags and should include a copy of the student registry with details of how to contact parents. It is especially important to include details of students’ special needs in the bag.

#### 3.2.2. More Supporting Staff Required to Support Evacuation Phase

The aim of evacuation while students are on campus is to follow the predefined evacuation route(s) and arrive at the assembly points. This focus on the evacuation procedure (the warning systems, the evacuation routes, and the actual evacuation process) required the research team to look at whether sufficient supportive staff were available to help students with disabilities to be evacuated.

Our research suggests that these staffing levels may not be sufficient to enable safe and swift evacuation in the event of an actual evacuation emergency. Students with different types of disabilities require different care and attention during the evacuation phase. If students with cerebral palsy have good cognitive level, they still need someone to help with the evacuation. If they do not have a wheelchair next to them, they each need two staff to support them at the same time. If students are sitting in a wheelchair, two teachers can carry the wheelchair downstairs. It is not easy to push the wheelchairs; if students are not very heavy teachers normally prefer to carry them downstairs. Students without disabilities or with only minor disabilities will act quickly; they can take action by themselves and are happy to ask peers to copy their behaviours. They are able to help with the evacuation procedures. Students with autism sometimes do not want to follow the instructions and/or procedures because they might be moody. When students with autism become moody, they may not want to work with people, they may want to run around. In these circumstances, teachers will need more support to ensure that the students evacuate to the right assembly points. Students might be too hyperactive and run in any direction and not remain in the group.

For example, teachers normally on the second floor were concerned about the evacuation because they were not sure if extra personnel and rescuers would arrive in time to help. In addition, children with more severe disabilities would need more time for evacuation. Participant A2 said, “In our previous exercise we have found that it was extremely difficult to evacuate level-4 students” (Participant A2). Students with physical or multiple disabilities require staff to help put on shoes and to carry wheelchairs. “It is more difficult for levels 3 and 4 students to evacuate at the same time without help” (Participant B2). “If an unscheduled alarm was issued, we do not want to evacuate those with wheelchairs because we were not sure if we would have anyone to come to help us” (Participant A3). Some teachers said levels 1 and 2 students have more issues because they can move freely so they might decide not to follow the instructions or run away; special education teachers with more experience would be better able to deal with these situations. “I noticed that children had gone missing during the evacuation process, so we need to assign more staff to support. Levels 3 and 4 students need staff to take them to use the accessible ramps; sometimes level-2 students need help from teachers when they support level-1 students” (B2).

“We will ask rescue teams to help if children really cannot take action by themselves or refuse to cooperate. We will do a head count and deal with those who do not arrive at the assembly point” (Participant B3). Carers and parents can be included to be trained to help with evacuation. “Some special education students have carers or parents in the class with them. They can be included in the support group; if we train them how to evacuate it could reduce the workload (burden) of teachers” (Participant B1).

It is suggested that emergency evacuation plans must deal separately with the challenges and resource requirements for each category of students with different types of disability. The ratio between staff and students with disabilities is crucial in the evacuation phase. Current guidance says that each special education class should have no more than 10 students enrolled [23]. Since the number of disabled students varied every year, normally these primary schools adapt the mixed-aged class, hence students from year 1 to 6 could study in the same class. The guidance also stipulates that each special education class should have two teachers, and one teaching assistant for every 15 disabled students. An extra part-time teaching assistant should be made available when the class has students with severe disabilities.

While all four cases had sufficient teachers and teaching assistants for normal situations, emergency evacuations might require additional resources, particularly to support students with more severe disabilities. Teachers from other classes could be assigned to support the evacuation process. It is possible to use more student assistants to support students with more severe disabilities, for example, Cases B and C. Level-1 students can remind level-2 students regarding the evacuation routes and assembly points.

The research group recommended considering the use of different types of supportive staff to help improve the situation where teaching resources are restricted. Administration staff, subject teachers, security guards, and clerks are external human resources to support the response and evacuation. Parents in attendance at special education classes can help students with rapid response and support evacuation. Identifying such resources, and providing appropriate training and preparation, should be included in the emergency plan, and particular attention must be paid to the need for coordination in lessons where students are not learning in their normal classrooms. Students will need help to return to the classrooms during the recovery phase. Carers should pay attention to the mental state of the students.

### 3.3. Earthquake Risk Reduction and Resilience Curriculum

The Ministry of Education has set up a risk reduction and resilience education curriculum for schools to follow, including teaching materials, supportive learning resources, and e-learning resources [28,29,30,31]. Other arrangements also include topics in natural hazards and disaster risk reduction, participating in earthquake simulator drills, field trips to visit disaster risk reduction museums, and earthquake response training and exercises. The Ministry of Education requires schools to conduct an earthquake exercise on the Annual DRR Day on 21 September and a multiple-hazard exercise in March. It is also recommended that parents participate in the exercises since earthquakes do not take place only in schools. This section summarised the feedback from interviewees regarding the improvement of the DRR curriculum.

#### 3.3.1. Clarify Learning Outcomes for Students with Different Levels and Severity of Disabilities

Although it is suggested to use the individual education programme to improve students’ knowledge and skills in response to disasters [13,16], interviewees have suggested that DRR education could be integrated into the formal curriculum and develop specific learning outcomes for students with different levels of disabilities (Participant A2, Participant A3). Schools could take into consideration individual students’ capacity and ability in the response, evacuation, and assembly phases, including physical environment and supportive staff arrangements (Participant B4).

Learning outcomes should identify the level of knowledge and practical skills (Participant B1, Participant B4). For example, level-1 and level-2 students are more capable of carrying out practice actions; hence the learning outcomes should use pictures, videos, and practical drills to increase their knowledge about earthquakes and methods for response. At the same time, they can also learn skills to help teachers to support level-3 and level-4 students (Participant A1, Participant A3, and Participant B4). Learning outcomes for level-3 and level-4 students should emphasise participation and increase their capacity for positive response in evacuation drills (Participant A1, Participant A2, and Participant A4), see Table 3.

#### 3.3.2. Using Pre-Exercise Training to Help Students with Disabilities to Respond Better to Earthquake Risks

Teachers in special education classes have recognised the importance of running earthquake response exercises to help students with disabilities acquire relevant skills to respond to potential hazards. “We have more frequent earthquakes, and the impact of the earthquakes is larger, so we have started to find the importance of participating in the earthquake response exercises. We like students to practice more often and to gain appropriate disaster risk knowledge” (Participant A3). Since students with disabilities have different needs and learning methods, it is suggested that attention should be paid to design and delivery of the exercises.

Most teachers agreed that students with disabilities should have received more repetitive training and lessons before the annual exercise. Students with disabilities require more time to understand the change of environment and to learn the response procedures, including familiarisation with the alarm sounds, response procedures, and evacuation routes (including maps) in different classrooms and playgrounds. These pre-exercise training sessions were “to instruct students to learn the methods and process of responding to earthquakes, so what we taught students beforehand will determine the success of the exercise” (Participant B4). At the same time, special classes could increase the frequency of exercises, so children are more familiar with the operation procedures, in particular level-1 and level-2 students, who are more able to look after themselves. This would allow teachers to provide more help to level-3 and level-4 students.

Participant B1 recommended that children with disabilities should be informed in advance about the exercise to avoid panic and stress, although the pre-exercise preparation reduced the realism of the exercise. For example, “Level-4 students were very nervous when the alarm sounded. After explanations several times, they started to get used to it” (Participant A2). Teachers also reminded students about the exercise on the day when the exercise was going to take place “because these students are forgettable” (Participant B3). It is recommended that more and regular training and exercises should be included in the curriculum for special education class students.

#### 3.3.3. Enhancing Competences and Capacities through Training the Trainers

Most of the teachers have obtained knowledge about earthquake response; however, there was still divergence. Teachers and teaching assistants of special education classes should increase their knowledge of earthquake hazards and response techniques (Participant A1, Participant B1), and familiarise themselves with students’ needs and requirements (Participant A2, Participant B2); in addition, teachers should understand how to maximise the safety of both students and themselves.

Teachers are the main carers of students with disabilities while in the school; it is also recognised that teachers work with support staff and relevant helpers, including parents, carers, social workers, psychologists, and first responders, to be able to provide appropriate help to support students with disabilities. In addition to general disaster risk reduction knowledge, it is suggested that both teachers and support staff should learn the following points to make sure that they meet the requirements of students with disabilities:

Impact of the earthquake in the classrooms: location of safe areas and available protection devices and equipment.Correct earthquake preparedness and response measures: since students with different types and severity of disabilities require different support, teachers and support staff should identify individual needs and make relevant emergency plans for these students.Decision-making capacity: in case of lack of support staff and/or unexpected situation, for example students not cooperating, teachers and staff need to be adaptive and flexible in response.

## 4. Conclusions

The research focused on four schools in Taipei city that include special education classes to investigate the effectiveness of their disaster risk reduction arrangements. Challenges and issues were identified. The research team summarised the recommendations from interviewees to provide suggestion for policy change.

First, priorities of safe campus physical environment were identified. In the area of school disaster management plans, the ratio between staff and students with disabilities should be paid attention. The individualised plan should be adapted for planning so that sufficient support could be gained, in particular in the evacuation phase. In the staff support, it is also recommended that special education teachers not only learn relevant knowledge and skills about disaster risk reduction, but they should also learn the skills of how to support their students. In addition, internal and external supporting staff, for example administrators, and parents should be involved in helping with the response and evacuation. In the risk reduction and resilient curriculum perspective, the research team recommended recognising the individual education plan and establishing clear learning outcomes for students with different severity of disabilities to facilitate more effective learning experience, including knowledge and skills.

Although the research only used four general schools in Taipei city that include special education classes as examples to explore the challenges, recommendations could be shared with other schools with similar settings. It is suggested that the arrangements for students with disabilities in schools located in rural areas or high-risk areas should be studied separately. In addition, investigation should be performed in normal classes that have students with minor disabilities. It should also be noted that the research focused on the opinions of schoolteachers and support staff; a separate study asking parents, carers, and students themselves about their challenges and needs during earthquake responses may well identify other areas for improvement.

## Figures and Tables

**Figure 1 ijerph-19-08750-f001:**
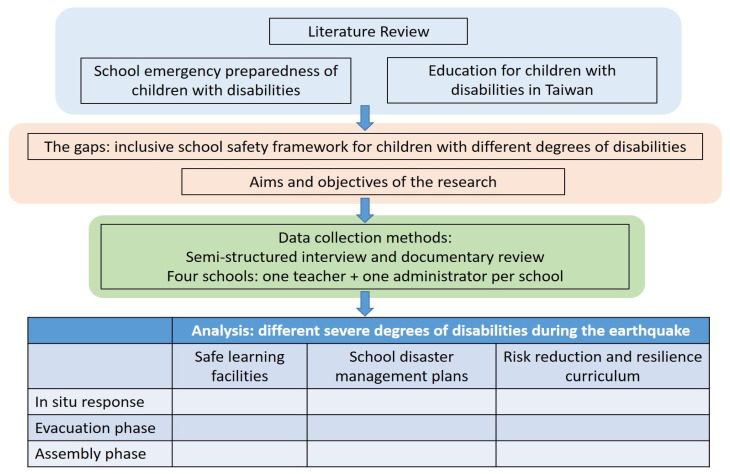
Process of research and framework of analysis.

**Table 1 ijerph-19-08750-t001:** Challenges and requirements of school infrastructure for students with different types of disabilities.

Types of Disabilities	Challenges for Responding to Emergencies	School Infrastructure Requirements	Type of Support Need for Evacuation
Developmental disabilities (intellectual disability)	Insufficient recognition to risk. Slow in physical movement and reaction.	Simple and easy memorable signals.	Emergency escape equipment installation. Support those who need help for evacuation. Effective information dissemination.
Visual impairments	Unable to differentiate shapes of objects, narrow vision, low optical ability, not easy to cognise colours. Rely on white cane (sticks) and supportive tools.	Conspicuous signs. Audio signals. Reduce the level of floors. Enhanced accessible equipment. Increase space for using canes. Simplified evacuation route.	
Hearing Impairments	Trouble hearing consonants, difficulty understand words, noise in the ear. Difficult to receive message/voice/signals.	Using visual signs and signals. Vibrant alarms.	Effective information dissemination.
Physical	Upper or lower limb loss or disability, manual dexterity, and disability in coordination with different organs of the body. Rely on assistive devices of mobility aids (crutches, canes wheelchairs and artificial limbs) to obtain mobility.	Reduce the level of floors. Increase space for wheelchairs and canes. Simplified evacuation route. Provide supportive equipment. Easy to operate equipment.	Emergency escape equipment installation. Support those who need help for evacuation. Effective information dissemination.
Multiple disabilities	Simultaneous occurrence of two or more disabling conditions that affect learning or other important life functions.	Simple and easy memorable signals. Reduce the level of floors. Increase space for wheelchairs and canes. Simplified evacuation route. Provide supportive equipment. Easy to operate equipment.	Emergency escape equipment installation. Support those who need help for evacuation. Effective information dissemination.

Modified from [15].

**Table 2 ijerph-19-08750-t002:** Participant information.

Status of Special Education Class	Participant Schools			
	Case A	Case B	Case C	Case D
Total number of the classes	1.5	4	4	1.5
Total number of students with disabilities	9	28	27	11
Number of students with level-1 disabilities	2	7	6	2
Number of students with level-2 disabilities	1	11	8	5
Number of students with level-3 disabilities	4	4	12	3
Number of students with level-4 disabilities	2	6	1	1
Number of teachers	3	8	8	3
Number of teaching assistants	1	1	2	1
Number of student assistants	1	2	2	1
Teaching classroom floor	1 *	1	2	1
PE classroom floor	1	1	2	B1/1
Daily skill classroom floor	1	1	2	1
Evacuation route	Straight-line distance to the assembly point.	Some students go via stairs; some evacuate via ramp.	Go through corridor, then downstairs to the assembly point.	2 points gathering; meet up in the first refuge point then go to assembly point.
Distance of the assembly point	Less than 50 m		More than 50 m	
Location of the assembly point	Fixed location	Non-fixed location	Fixed location	Fixed location

* Floor 1 = Ground floor in the UK. Source: adapted from School disaster management plans.

**Table 3 ijerph-19-08750-t003:** Learning outcomes for students with different levels of disabilities.

Levels of Disability	Learning Outcomes for Earthquake Response
1	Knowledge: To be able to understand the earthquake hazard; to perceive earthquake risks; to recognise that it is not possible to predict earthquakes; to learn about using emergency grab bags and functions of risk communication platforms (messengers); to be able to discuss with family members how to respond and evacuate. Practical skills: To be able to precisely complete the evacuation procedure; to be able to protect one’s head and to be aware of falling objects; to be able to reach the assembly point on their own; to be familiar with different evacuation routes from different classrooms; to be able to help teachers to verbally guide level-2 students during evacuation and to reach the assembly point.
2	Knowledge: to be able to identify earthquake pictures and videos; to be able to distinguish earthquake warning/alert/siren; to learn to follow peers to meeting points. Practical skills: to be able to do hide, drop, and hold on with reminders; to be able to protect their heads and able to follow teachers/carers/peers to reach meeting points.
3	Knowledge: to be able to recognise the earthquake siren and not be panicked by the alarm. Skill: to be able to participate in earthquake drills in different locations; can be evacuated with others’ support.
4	Knowledge: To be able to seek help verbally if they have verbal skills. Practical skills: if they have the manual skills, to lock the wheelchair, hold things, and cover heads. Students with low cognitive skills should participate in the earthquake exercise in different venues and participate in evacuation and response.

## Data Availability

The research project has received the research ethics approval from Ming Chuan University. All participants were giving a brief about the project before data collection and signed the consent form.

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
