# Peer review of "Earthquake Response for Students with Different Severe Degrees of Disabilities: An Investigation of the Special Education Classes in Primary Schools in Taipei"

_ijerph, 2022, doi:10.3390/ijerph19148750_

Round 1
Reviewer 1 Report
Dear authors,
Thank you first for your attention to the topic of disaster response for students with disabilities in Taiwan, China.
This study investigated the challenges of earthquake preparation and response for students with different severe degrees of disabilities who enroll in the special education classes in general primary schools. This is a very important research topic. The paper also gave some interesting opinions.
However, from the perspective of an academic paper, there is still much space for progress in sample selection and quantitative analysis. Specifically, this study mainly focused on qualitative analysis. Although the semi-structured questionnaire was used, the sample size is too small and only one teachers and one school administrator in each school are interviewed. The main body should be the students themselves. Thus, the representativeness of the results needs to be further tested.
It is suggested that it should strengthen the application of quantitative analysis methods, and enhance the corroboration of objective data, to improve the objectivity and representativeness of the results.
In addition, Taiwan is not a sovereign country. It must be clearly revised in the first sentence of the article.
It is recommended that the samples should be supplemented and the quantitative analysis should be strengthened. After that, it could be returned for review.
Reviewer 2 Report
In Table 1, the number of children with the given type / degree of disability is not clear. It isn't known which step is the lightest and heaviest.
Reviewer 3 Report
Title: Is it possible to rephrase the title and make it shorter. Mentioning the research site (Taiwan) in the title is also recommended.
Abstract: The abstract needs to be rewritten with some emphasis on the below points:
Problem statement (lines 14-15): As mentioned at the beginning of the abstract, the comprehensive school safety framework and special education law 2014 - are different? If so, it would be important to rephrase the sentences and sharpen the focus of the work. If this is a policy review, it is important to declare that your paper would look into policies associated with school safety for diverse disabilities, for example. I understand that your main argument is to showcase the inefficiency of existing evacuation policy and practice in cases of inclusive education (as the current policy infrastructure accommodates only specialised schools, not diverse disabilities in inclusive schools).
Objectives of the research (lines 21-25): In the backdrop of the earlier comments on the title and the problem statement, the three objectives are extremely unclear. It appears from the objectives that the research wants to examine 1) the challenges of disability situations in regular time; to compare with 2) the challenges in is earthquake evacuation situations; to propose 3) policy recommendations. It seems very wide arching and over ambitious to capture in one article.
Methodology (line 26-26): The information on methodology in the abstract is inadequate. In one single statement, the authors only mentioned two techniques implemented without giving any further clue of with who.
Findings (line 27-31): the three points in the finding do not categorically address the points set forward through the declaration of the objectives. Either the objectives need to be rephrased to sync with the findings or the other way around. But, they need to be matched.
This abstract has touched upon four fundamental points of research, where is the problem, what would be achieved, how to achieve and what has been achieved. However, abstracts also accommodate information on the novelty, audiences and implication of particular research, which is missing here. Adding this information might significantly improve the quality of the abstract by giving the readers a signal of what to expect.
Introduction:
Line 36: “Taiwan is has very frequent earthquakes” – please check the sentence construction.
Right after the 1st paragraph (with some information on the intensity and magnitude of earthquakes in Taiwan) the discussion directly moved to the policy infrastructure without setting a scene of how earthquakes have impacted children during school hours in general and diverse disabled children in particular. A scene setter is very important here.
1.1 Disaster resilience of children with disabilities (line 75): I do not think that the whole section has done any justice to the discussion on ‘disaster resilience’, rather it describes the disaster preparedness status quo and its challenges. The discussion also seems isolated (not properly synchronised) from the mainstream discussion aligning to the research objectives. In the later discussion, the Comprehensive School Safety Framework (UNDRR) was found to be adopted as the cannon of ‘disaster resilience’. In case, the paper wishes to have a discussion on disaster resilience for schools, that discussion on the framework and its indicators needs to come here.
1.2. Education for children with disabilities in Taiwan: this section identified different disability conditions and linked them to two different education systems. This identifies several categories of disabilities which constitutes 95% of the total disabled category are going to the mainstream inclusive education systems. For a robust theoretical framework, it would have been interesting to expand the debate on how those disabilities are so diverse when it comes to earthquake response and what kind of capacity a school infrastructure needs to have to perform better evacuation. As the objectives of the research set out to evaluate those capacities (a list of indicators which constitute those capacities) for the schools with inclusive education.
Methodology: This is apparently one of the weakest sections of this paper. This grossly mentions semi-structured interviews and archival reviews, without giving much detail about how those exercises will be performed. I have not seen any discussion on ethical considerations, which is very important in such research. In-depth interviews with teachers and parents on experiences of challenges during earthquakes would be very useful. Also, how variabilities of disability conditions were captured in the data collection, and so on is missing in the methodology discussion. In such research, speaking to relevant government departments and ministries and understanding their positions, whether they recognise their limitation if they have any plans to overcome those challenges, and so on are very important discussions, which assumingly have not taken place. Since the research has adopted the UNDRR’s school safety framework, it is important to discuss the key components/indicators of safety/resilience in the framework and how that was evaluated in the study. The figure of the research process and framework does not add any value to the whole discussion. It just occupies some extra space.
Reference: In cases of cited references the date of assessing the article has been mentioned which seems extremely unusual. In cases of website references (not for journal articles), we mention the date of access, as website information could change. But I have never seen any academic article mentioning the date of assessment.
Overall, the paper has significant flaws in the theoretical framework and methodology. Therefore, I didn't feel it is important to read the results and finding until the theoretical framework and mythologies are amended.
Round 2
Reviewer 1 Report
First of all, I would like to thank the authors for their explanations of the first review comments.
I have also noticed the efforts and attempts made by the authors for this research. However, I'm sorry that the author's explanation of the review comments is not convincing. To be exact, the authors have no plan to take further action for improving the contents and methods mentioned in last review comments. Personally, I still believe that the derivation of this research conclusion is incomplete due to the lack of direct evidence or objective historical data from the research subject (disabled students).
Therefore, I regret that I cannot agree with the author's supplementary explanation for the first review comments.
In the current research state, I do not recommend this article to be published.
Author Response
Dear the Reviewer
It is extremely grateful to the positive comment. The derivation of this research conclusion have been confirmed carefully due to the characteristics of students with disabilities.
In the response to the first review, we have explained that “it is challengeable to interview students with disabilities; their disability background has made them more vulnerable to interview formats. To comply with the research ethics guidance, the research team has not interviewed students.” In addition, due the nature of disabilities, most students were not easy to express their opinions independently, normally carers and teachers interpret students’ opinions. It was decided that we only interview teachers and administrators in the schools.
We hope the supplementary explanations of the second review that really help the students with disabilities enrolling in normal schools for the challenges of earthquake preparation and response.
Thank you very much.
Best regards,